# The State of Experimental Research on Community Interventions to Reduce Greenhouse Gas Emissions—A Systematic Review

**Anthony Biglan [1], Andrew C. Bonner [2], Magnus Johansson [3],\* 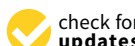, Jessica L. Ghai [4],**
**Mark J. Van Ryzin [1], Tiffany L. Dubuc [5], Holly A. Seniuk [6], Julia H. Fiebig [7] and Lisa W. Coyne [8]**

[1]  Oregon Research Institute, Eugene, OR 97403, USA; tony@ori.org (A.B.); markv@ori.org (M.J.V.R.)
[2]  Department of Psychology, University of Florida, Gainesville, FL 32601, USA; acbonner@ufl.edu
[3]  Department of Behavioural Science, Oslo Metropolitan University, NO-0130 Oslo, Norway
[4]  Wheelock College of Education and Human Development, Boston University, Boston, MA 02215, USA; jessica.morton1@gmail.com
[5]  Department of Psychology, University of Nevada, Reno, NV 89557, USA; tiffany.dubuc.bcba@gmail.com
[6]  Behavior Analyst Certification Board®, Littleton, CO 80127, USA; holly@bacb.com
[7]  Department of Special Education, Ball State University, Muncie, IN 47306, USA; jhfiebig@gmail.com
[8]  Department of Psychiatry, Harvard Medical School, Boston, MA 02115, USA; dr.lisacoyne@gmail.com
\*  Correspondence: magjoh@oslomet.no

**Abstract:** This paper reviews research on community efforts to reduce greenhouse gas emissions. We conducted a systematic search of the relevant literature, and supplemented our findings with an analysis of review papers previously published on the topic. The results indicate that there have been no peer-reviewed experimental evaluations of community-wide interventions to reduce greenhouse gases involving electricity, refrigeration, or food. The lack of findings limits the conclusions which can be made about the efficacy of these efforts. As a result, we are not accumulating effective interventions, and some communities may be implementing strategies that are not effective. We advocate for the funding of experimental evaluations of multi-sector community interventions to reduce greenhouse gas emissions. Such interventions would attempt to engage every sector of the community in identifying and implementing policies and practices to reduce emissions. Comprehensive multi-sector interventions are likely to have synergistic effects, such that the total impact is greater than the sum of the impact of the individual components. We describe the value of interrupted time-series designs as an alternative to randomized trials, because these designs confer particular advantages for the evaluation of strategies in entire communities.

**Keywords:** systematic review; community intervention; greenhouse gas emissions; climate change

---

## 1. Introduction

This paper presents a systematic review of research on community interventions to reduce greenhouse gas emissions, with particular attention being paid to experimental evaluations of these interventions. The threats posed by climate change are well documented. Indeed, there is mounting evidence that earlier predictions of the impacts of global warming consistently underestimated the extent and seriousness of the damages [1,2]. Furthermore, the rate of greenhouse gas (GHG) emissions is increasing; in 2018, the rate of GHG emissions reached an all-time high [3], following yearly increases since the 1980s. For this reason, the Coalition of Behavioral Science Organizations created a Task Force on Climate Change to examine the state of research on reducing greenhouse gas emissions.

We hope to identify the ways in which behavioral science can make a more significant contribution to reducing emissions.

Community interventions are one viable path forward to reducing emissions. Experimental evaluations of multi-sector community interventions have been conducted to address a variety of psychological, behavioral, and health problems. Studies have tested whether cardiovascular disease could be reduced in the entire population of a community [4,5]. The National Cancer Institute funded a randomized controlled trial conducted in eleven matched pairs of communities which tested whether the prevalence of smoking could be reduced through a multi-sector community-wide campaign [6]. Research on the prevention of adolescent problems has used randomized trials to evaluate interventions to prevent smoking [7], other substance use [8], and substance use and delinquency [9] in entire communities. All of these interventions involved organizing multiple sectors of communities in small to moderately-sized communities (populations of 2000 to 125,000) to implement multiple strategies for affecting the targeted outcomes. Although the studies focused on adult health had a limited impact, the studies on preventing youth problems all had beneficial effects [10].

In recent years, comprehensive multi-sector community interventions have typically followed a collective impact strategy [11]. The critical features of such interventions include: (a) the identification of all of the sectors of the community that could, if mobilized, influence the targeted outcome; (b) the organization of multiple sectors to work collaboratively to achieve the targeted outcome (e.g., the cessation of smoking, a reduction in cardiovascular risk, the prevention of youth smoking); (c) the identification of specific things that each sector can do to contribute to goal achievement; and (d) the monitoring and support of each sector's efforts by a 'backbone' organization [11]. The sectors of the community that are involved depend on the goal of the effort. For example, in health-related interventions, the sectors typically include healthcare providers, schools, human service organizations, businesses, and local government.

Multi-sector interventions have the potential to create synergistic effects because changes in any one sector—for example, businesses adopting industrial GHG reduction policies or schools teaching students about the need to reduce GHG emissions—could influence other sectors, such as households. Although there has been a fair amount of research on the reduction of individual and household emissions of greenhouse gases [12–16], less evidence exists regarding the impact of community interventions—especially those that mobilize multiple sectors of the community.

Certainly, community interventions are not the only strategy through which emissions can be reduced. For example, national policies to increase the cost of emissions have the potential to reduce emissions [17,18], and experimental research evaluating strategies for getting such policies adopted is badly needed. Absent a strong and widespread governmental commitment to such policies, however, community interventions may represent the most readily accessible tool to address climate change on a global level.

Thus, the present review sought to analyze the extent of the literature on community-based interventions that target the reduction of GHG emissions and make use of experimental research designs. Our goal was to identify the most promising strategies so that further research can build on existing evidence by: (1) strengthening the effectiveness of strategies showing positive effects; and (2) scaling up the best strategies so that they can be employed in communities worldwide.

We focused on experimental evaluations of community interventions for three reasons. First, experimental methods provide the most efficient and accurate way of determining the efficacy of an intervention. Despite the fact that there are many efforts worldwide to reduce greenhouse gas emissions in communities [19–25], it is unclear how effective these efforts are and which strategies are most effective. Without precise information about the impact of intervention strategies, it is impossible to know which strategies should be widely implemented and could be adapted to other settings. In the absence of a robust process of experimental evaluation, numerous communities may expend valuable resources implementing strategies that fail or have minimal impact. Furthermore, research demonstrating that a particular strategy has a reliable impact on emissions provides a basis for all

further community interventions to build upon that strategy, and for prompting the development of policy that supports the dissemination of the strategy.

Second, experimental evaluation enables incremental improvement in effectiveness. In an evolutionary process of variation and selection, strategies are tested, and those that have the greatest effect are retained. Those that fail to have an impact are abandoned or modified. With experimental evaluation, we have the possibility of identifying promising interventions that can be further strengthened by testing innovative variations of the intervention. In essence, the routine use of experimental evaluation will yield increasingly powerful strategies that have the potential to accelerate reductions in GHG emissions. This view is supported by the extensive progress that experimental research has made possible in medicine [26], clinical psychology [27], prevention science [28], and other areas of behavioral science [29].

Third, strategies that are empirically demonstrated to be effective and are published in the literature become available to communities, and, if adopted, will contribute to accelerating progress in reducing emissions. In the absence of such evidence, communities are more likely to continue to use strategies that are less effective than they could be, or may even be counterproductive. Arguably, policy and practice informed by robust science could be our most powerful tool.

We reasoned that experimental evaluations of community interventions would provide the best guidance to communities that are striving to reduce GHG emissions. The objective was to find peer-reviewed experimental evaluations so that the research and practice communities could make use of the most successful strategies, and could build on them via further refinement and experimentation.

## 2. Method

### 2.1. Eligibility Criteria

This review was organized to identify experimental research on community interventions aiming to reduce GHG emissions. We limited the search to studies targeting the three areas most likely to have the largest impact on GHG emissions, based on Hawken's [30] Project Drawdown (https://drawdown.org/solutions). These are described in more detail in Table 1.

**Table 1.** Exact search terms used with the Scopus database.

| |
|---|
| TITLE-ABS-KEY(community OR communities) AND |
| TITLE-ABS-KEY("climate change" OR "global warm*" OR "greenhouse gas*" OR ghg OR "carbon emission*" OR "co2 emission*") AND |
| TITLE-ABS-KEY(trial* OR random* OR "interrupted time-series" OR "multiple baseline" OR "time-series design" OR "experiment*" OR single-case OR interven*) AND |
| TITLE-ABS-KEY(energy OR electricity OR food OR plant-based OR diet OR refriger* OR cool* OR chlorofluorocarbon OR cfc OR cryogenic* OR "heat remov*" OR "heat recov*" OR "heat exchange") AND |
| (Exclude(subjarea, "chem") or exclude(subjarea, "ceng") or exclude(subjarea, "phys") or exclude (subjarea, "mate")) and (exclude (subjarea, "bioc") or exclude(subjarea, "medi") or exclude(subjarea, "comp") or exclude(subjarea, "math") or exclude(subjarea, "immu") or exclude(subjarea, "nurs") or exclude(subjarea, "phar") or exclude(subjarea, "arts") or exclude(subjarea, "vete") or exclude(subjarea, "heal")) |

We defined a community intervention as an approach that (a) used interventions targeting multiple sectors of the community, and (b) was applied throughout an entire geopolitical entity no larger than a city (e.g., neighborhoods, villages, towns, or cities).

An example of a multi-sector community intervention might include a component targeting schools, to teach children about the importance of reducing emissions, a component involving the city council adopting ordinances, and a component targeting local businesses to assess and reduce their emissions. To qualify as a community intervention, the strategy was required to have targeted the entire community, and to have been characterized by the aforementioned definitive features.

We defined experimental evaluations as those using a group-based design, with at least one control and one intervention group or condition, or an interrupted time-series design.

### 2.2. Information Sources

Studies were identified for inclusion by conducting searches using the databases Web of Science and Scopus. These were selected because of their broad reach in the areas of social and behavioral science. The Scopus search was conducted on 25 July 2019, and the Web of Science search on 29 July 2019.

### 2.3. Search

The precise terms used in the search are shown in Table 1, using Scopus syntax. No limit on the publication year was used, and only peer-reviewed English language publications were searched. Additional articles were identified by reviewing the results from Gelino et al. [31], who conducted a systematic review of six behavior analytic journals for articles related to GHG emissions. Articles identified in their review were included in ours if they fit the inclusion criteria.

As shown in Table 1, titles, abstracts and keywords were searched for the word "community" or "communities", in combination with terms relevant to climate change and experimental research designs, and terms targeting the three highest impact areas. These were identified as being related to food/diet, energy/electricity, and refrigeration/cooling. Papers that focused on the physical sciences and other areas deemed unlikely to produce relevant results were filtered out. These areas were manually screened prior to exclusion, and are listed in the bottom section of Table 1.

### 2.4. Study Selection

We identified relevant articles across three stages of coding (see Appendix A for the full documentation of the coding stages). Two doctoral students in behavioral science oversaw the coding process and completed stage one coding. Before stage one coding began, a training module was created to increase reliability. A quasi-random set of 20 articles was selected and coded by trainees independently. The coders' records were separately compared to an expert consensus record using the block-by-block method [30] with three separate codes: (a) irrelevant, (b) relevant, and (c) a review and/or needed to read the entire article to code properly. Codes where observers agreed were treated as complete agreements, and codes where observers disagreed were treated as complete disagreements. To calculate the overall agreement, we divided the total number of agreements by the total number of agreements and disagreements for each article. The percentage of the agreement for each article was then averaged across all of the articles to produce an overall agreement percentage. We set the mastery criterion at 80% agreement, and the two coders' reliability coefficients during training equaled 90% and 100%, respectively. During stage one, a second trained observer coded 28% of all of the articles, and their reliability equaled 95% (range, 33% to 100%).

During stage one, coding was conducted based on titles and abstracts. We retained all of the articles that met one of the following criteria for the next stage of coding: (a) described an experimental evaluation of an intervention aimed at reducing GHG emissions (e.g., lowering electricity consumption or gas usage) using real-world data (not simulated or conducted in a lab setting); (b) a literature review or a meta-analysis of interventions to reduce emissions; or (c) seemed relevant but could not be determined based on the abstract, and the entire article needed to be read to determine eligibility for inclusion. Each paper could be coded as more than one category. For example, a paper that was a review but also required full-text reading to code properly would have been coded as such (in fact, this was the case for 21 papers). If an article did not satisfy any of the conditions mentioned above, it was coded as irrelevant.

For stage two, full copies of the remaining articles were obtained. During this stage, the type and features of the experimental design used; the primary dependent variables; whether behavior was measured directly, by observation or by self-report; the intervention components utilized; and the

overall impact of the intervention was coded for each article. Articles that did not contain an experimental evaluation of a community intervention were excluded.

During stage three, we obtained full-text articles coded as a systematic review or meta-analysis. The editors of this manuscript pointed out that we could enhance the comprehensiveness by including pivotal review articles that were not captured in our initial search. Four such review articles were included [14,16,32,33]. The reference section of each article was inspected to identify additional articles that satisfied the inclusion criteria previously described. Any additional articles identified were then submitted to a stage-two coding. A PRISMA checklist [34] is linked in Supplementary Materials.

*2.5. Data Extraction Process*

**Experimental design**. To evaluate the experimental rigor, the type of design utilized was classified. According to Shadish et al. [35], the most rigorous group designs are characterized by three critical features: (a) the observation of dependent variables before and after the application of an independent variable, (b) the presence of a no-intervention control group, and (c) random assignment. Kazdin [36] characterized the most rigorous interrupted time-series designs according to three critical features. First, the repeated measurement of dependent variables within each experimental condition (i.e., baseline and one or more intervention conditions). Second, there must be at least one opportunity to compare the level and slope of the time series between the baseline condition and an intervention condition. Third, there must be at least one opportunity to test the replicability of an intervention effect. These features were coded for each type of experimental design. A strong design of either type was defined as having all three features present; a weak design was missing at least one of these features. The community size for each study was also noted.

**Dependent variables**. To evaluate the primary dependent variables, the data collection methods were analyzed first. Objective data were defined as being collected directly, or by observation if records were produced automatically (e.g., electricity consumption reported by utility) or by an independent observer (e.g., inspecting a consumer's natural gas meter). Subjective data were defined as being collected indirectly (e.g., by surveys and interviews). Next, the nature of the dependent variables was characterized, such as food waste or electricity consumption, and the units (e.g., kWh) were coded.

**Intervention components.** The intervention components were coded in detail and grouped thematically. Next, the interventions were cast broadly as antecedent-based and/or consequent-based. Antecedent-based interventions were defined as being those involving manipulations that occurred before behavior was emitted (e.g., antecedent information, social marketing campaigns, prompting). Consequent-based interventions were defined as being those involving manipulations that occurred after behavior occurred (e.g., incentives, performance feedback). Finally, we determined the differential effectiveness of the overall intervention package based on inferential statistics presented in text, or visual analysis based on descriptive statistics.

## 3. Results

Figure 1 presents the results of our search. A total of 1883 papers were identified from the two databases, with an additional 19 articles being identified from the Gelino et al. [31] review. The removal of 454 duplicates yielded 1448 papers. Of these, 1226 papers were removed based on the irrelevance of their title or abstract, while an additional 104 review papers were set aside for later analysis, resulting in an initial yield of 118 papers. In total, 94 of the 118 papers required full-text reading to code correctly; none of these contained a relevant evaluation of a multi-sector community intervention. Finally, 24 papers were coded as containing a relevant evaluation, but none contained an experimental evaluation of a multi-sector community intervention. The reference sections of the 104 review papers (plus the additional review papers identified during the peer-review process) were scanned for relevant articles. This snowball sampling yielded no studies that experimentally evaluated a multi-sector intervention in an entire community. Thus, the results of our search procedures yielded no studies of an

experimental evaluation of a multi-sector community intervention aimed at reducing GHG emissions involving the three target areas.

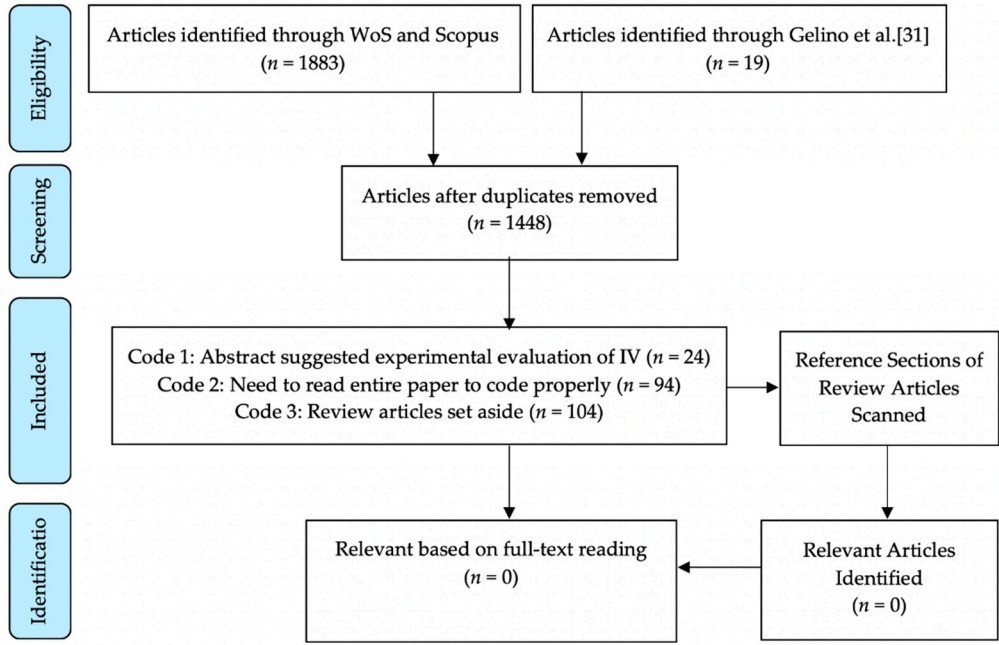

**Figure 1.** Flow chart of article inclusion and exclusion by stage.

We defined a community intervention as an approach that (a) organized multiple sectors of the community, and (b) was applied throughout an entire geopolitical entity no larger than a city (e.g., neighborhoods, villages, towns, or cities). Despite finding no relevant studies that satisfied this definition, we found twelve studies that contained an approach applied throughout an entire geopolitical entity. However, the vast majority of these studies did not target communities per se, but rather residents across whole states [37], multiple states [38,39], or entire countries [40,41]. In other words, the geopolitical entity was larger than a city. Two studies [42,43] contained an approach targeting a population of an appropriate size, but neither study included a multi-sector approach, and therefore they did not satisfy our definition of a community intervention.

## 4. Discussion

The most important conclusion drawn from the present analysis is that there is too little experimental research testing the impact of community strategies for affecting GHG emissions. Despite a comprehensive search, we did not find a single study that experimentally evaluated a multi-sector community-wide strategy for reducing GHG emissions in an entire community. Given the number and variety of community interventions that are being adopted worldwide [20,21,23,25], this result indicates a significant missed opportunity to identify and accumulate increasingly effective strategies.

The dearth of experimental evaluations is not due to a lack of community interventions to affect emissions. Indeed, the report of the European Network for Community-Led Initiatives On Climate Change and Sustainability [21] indicated that "the scope and diversity of community-led action on sustainability and climate change in Europe, while unknown, is vast." Similarly, in the U.S., at least 392 mayors [44] have joined in an effort to reduce emissions in their communities [44].

We are not arguing that the many community interventions that are underway are lacking in results. Landholm et al.'s [23] report on 38 community-based interventions in Europe notes that the intervening organizations indicated that " . . . energy generation through renewable sources, changes in personal transportation, and dietary change . . . reduced carbon footprint by 24%, 11%, and 7%,

respectively." However, the direct measures and designs used to produce these conclusions were not reported. Still, this indicates that some communities are making progress.

We believe that the effectiveness of these interventions and their components could be significantly increased through experimental evaluation. This is not to say that experimental evaluation will necessarily result in successful outcomes. Rather, we are saying that, over time and multiple studies, we will be able to retain interventions that have positive effects and eliminate or modify strategies that experimentation has shown to have no benefit.

Of particular concern is the absence of experimental evaluations of multi-sector interventions in which many or all of the sectors of a community are organized to work collaboratively on reducing the many ways in which a community contributes to GHG emissions [45]. We believe that such multi-sector efforts have the greatest potential to produce substantial and lasting reductions in emissions, thanks to the synergistic effects that appear likely when different sectors interact and work together. A multisector approach is the hallmark of community interventions, and is crucial for producing measurable changes in communities. In many of the twelve 'almost' relevant studies, several intervened on households by providing performance feedback or a free household energy audit. However, households represent just one sector of the community. Any intervention targeting households at the community level stands to be more effective if other areas/sectors of the community are also activated. For example, an intervention that includes a school-based education program about household energy use and GHG emissions may work in synergy with household interventions. Moreover, intervention components that activate other sectors—like local industry and businesses—stand to strengthen the intervention's impact, because household residents spend time (either working or patronizing) in these community sectors, representing additional pathways and opportunities for intervention.

We recognize that concepts such as Real-world Labs [46], Sustainable LivingLabs [47] and Urban Transition Labs [48] include several ambitions that are in line with our reasoning on multi-sector community research, including the importance of transdisciplinarity, local involvement and science–practice integration [49]. However, despite the frequent use of the word "experiment" in the referenced articles, none of them refer to experimental designs to evaluate the effects of the interventions.

The use of experimental evaluations in the human sciences has yielded an enormous body of evidence that is relevant to improving human well-being. Specifically, through the application of experimental methods, tested and effective interventions have been developed to address a wide range of problems. In clinical psychology, efficacious interventions have been evaluated and refined for the treatment of the most common and costly psychological and behavioral problems, including depression, anxiety, physical inactivity, obesity, antisocial behavior, and substance use disorders [50]. Similarly, prevention scientists using experimental evaluations have developed family and school interventions that have proven benefits in preventing the development of all of the most common and costly problems of childhood and adolescence, including depression, anxiety, academic failure, antisocial behavior, and substance use [28,51–56]. A wide variety of other fields have also embraced experimental methods, including medicine [57], political science [44,58], economics [45], and public policy [59–62]. We submit that similar progress will occur in the field of climate change if greater use is made of these methods. In what follows, we discuss the experimental methods that we believe are most likely to accelerate the ability of communities to reduce GHG emissions, and highlight relevant examples.

One of the most surprising things in our review of the literature is the number of papers that describe community interventions in multiple communities, but do not provide empirical evidence of the impact of the interventions on greenhouse gas emissions. There are qualitative case studies of various community interventions, with no information about their impact [63,64]. Some rely on reports of the intervention organization's impact, but do not indicate how the measures were obtained. Others provide qualitative analyses of the types of interventions being tried, but do not report on the

impact of the interventions [25]. Thus, they provide little guidance to communities that are seeking to implement the most effective strategies.

### 4.1. Limitations

One limitation of our analysis is that it was restricted to only three areas for the reduction of GHG emissions—refrigeration, energy generation, and food waste. It is possible that our search strategy missed experimental evaluations of community interventions focused on other sources of emissions. However, if this is the case, it raises the question of why there is a dearth of research on the highest impact areas. It is also possible that we missed community interventions focused on policy adoption, although an ongoing search of that literature has thus far failed to reveal such studies.

Another limitation was only including studies mentioning the word 'community' or 'communities' in the title, abstract or keywords. There may exist studies that otherwise fulfill our eligibility criteria that were not found due to this.

### 4.2. Experimental Methods

Given the paucity of experimental evaluations that our literature review has documented, there exists little understanding of the most effective ways to influence the climate-related behavior of individuals, households, and organizations. Pinpointing powerful functional relationships is foundational for developing interventions that can then be scaled up to affect behavior in entire communities.

The most widely used and best understood experimental method is the randomized controlled trial. There is, however, another form of experimental design which is less widely used, but is likely to be more efficient when it comes to evaluating community interventions. It is variously referred to as a Single-Case Design or an Interrupted Time-Series Design [65]. There is already a body of research showing the value of these designs for identifying interventions that affect environmentally-relevant behaviors [31].

Interrupted time-series designs involve the application of an independent variable to an outcome that is repeatedly measured over time (i.e., a time-series) [65]. The two most common single-case designs are the reversal design and the multiple baseline design (sometimes referred to as a stepped wedge design [66]). An example of a reversal design on electricity consumption was reported by Kohlenberg et al. [67]. They examined the impact of feedback and incentives on the use of electricity during peak hours across three families. They compared the usage during a series of two-week phases. In the first, the baseline phase, electricity use during peak hours was simply monitored. In the second phase, the families were given information about the need to reduce usage during peak hours. In the third phase, the families received feedback in the form of a light that turned on if their use exceeded 90% of peak levels. The fourth phase was a return to the baseline condition, when no information or feedback was given. In the fifth phase, the families were given feedback and a monetary incentive if they could reduce their peak rate by 50% or more. Finally, in the sixth phase, they returned to a baseline condition. Figure 2 presents the results of this study. The data consist of cumulative records in which each day's consumption was added to the previous day in each two-week phase. Thus, a decline in use is shown by a line with a lower total use over two weeks. As can be seen, the feedback diminished the use of electricity, and the addition of incentives produced a greater impact.

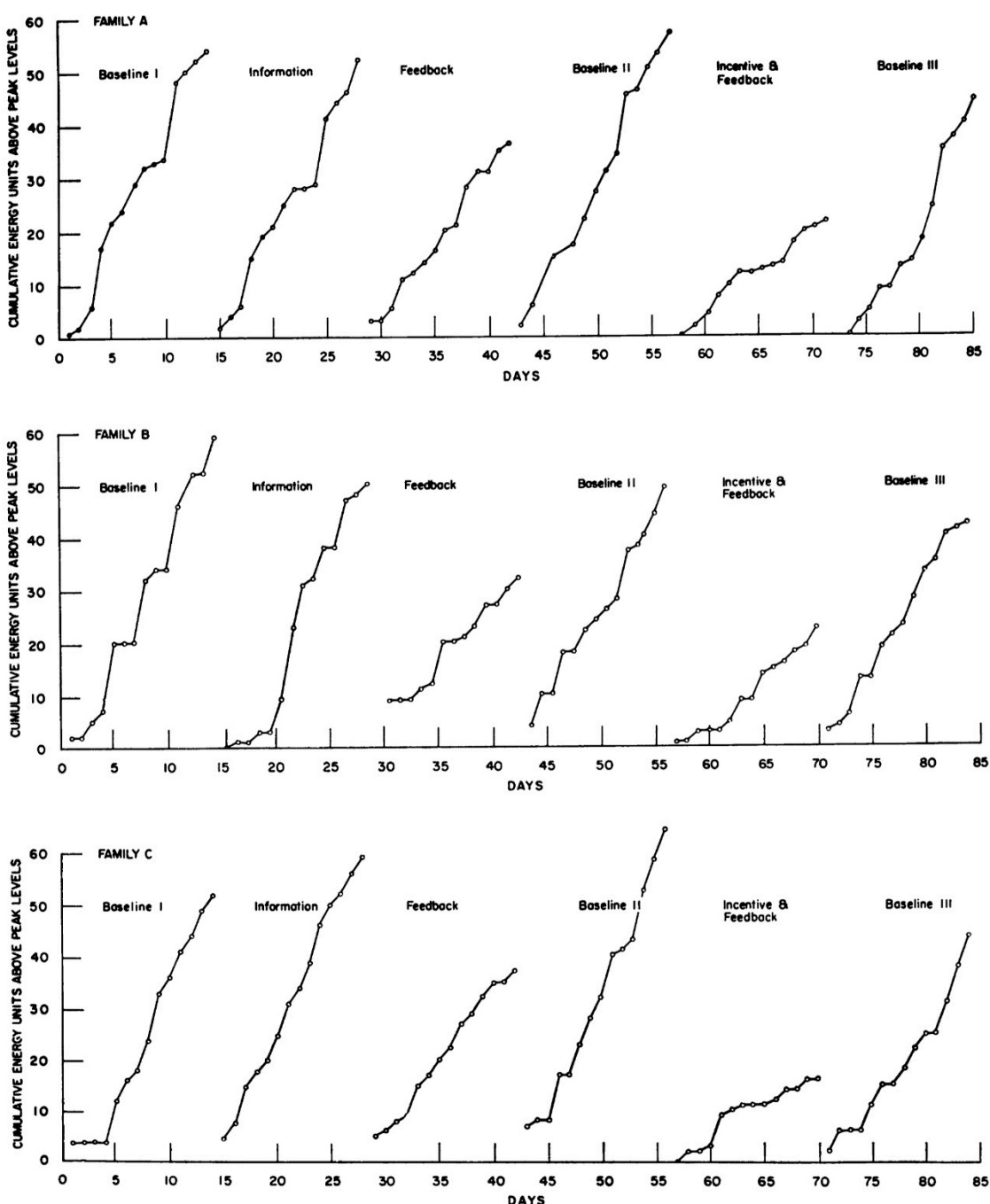

**Figure 2.** A reversal design showing cumulative electricity consumption, reported by Kohlenberg et al. [67] (Figure 2 © John Wiley and Sons. Reuse not permitted).

A study by Clayton and Nesnidol [68] employed a multiple baseline design to evaluate a strategy for reducing electricity consumption in a university classroom building. Specifically, they sought to have the lights turned off in classrooms at the end of the day through the use of a visual prompt by the light switch, which reminded people to turn off the lights and gave feedback about the percentage of classrooms in which the lights were turned off. For this study, the six-story building was divided into four sets of floors. As can be seen from Figure 3, the intervention increased the percentage of classrooms where the lights were turned off, and also reduced the variability in the percentage of classrooms that had them turned off. The evidence that it was the intervention that led to these changes comes both from the fact that the change in the percentage of classrooms with the lights off increased

when the intervention was implemented, and the fact that the percentage did not change for the floors where the intervention had not yet been implemented.

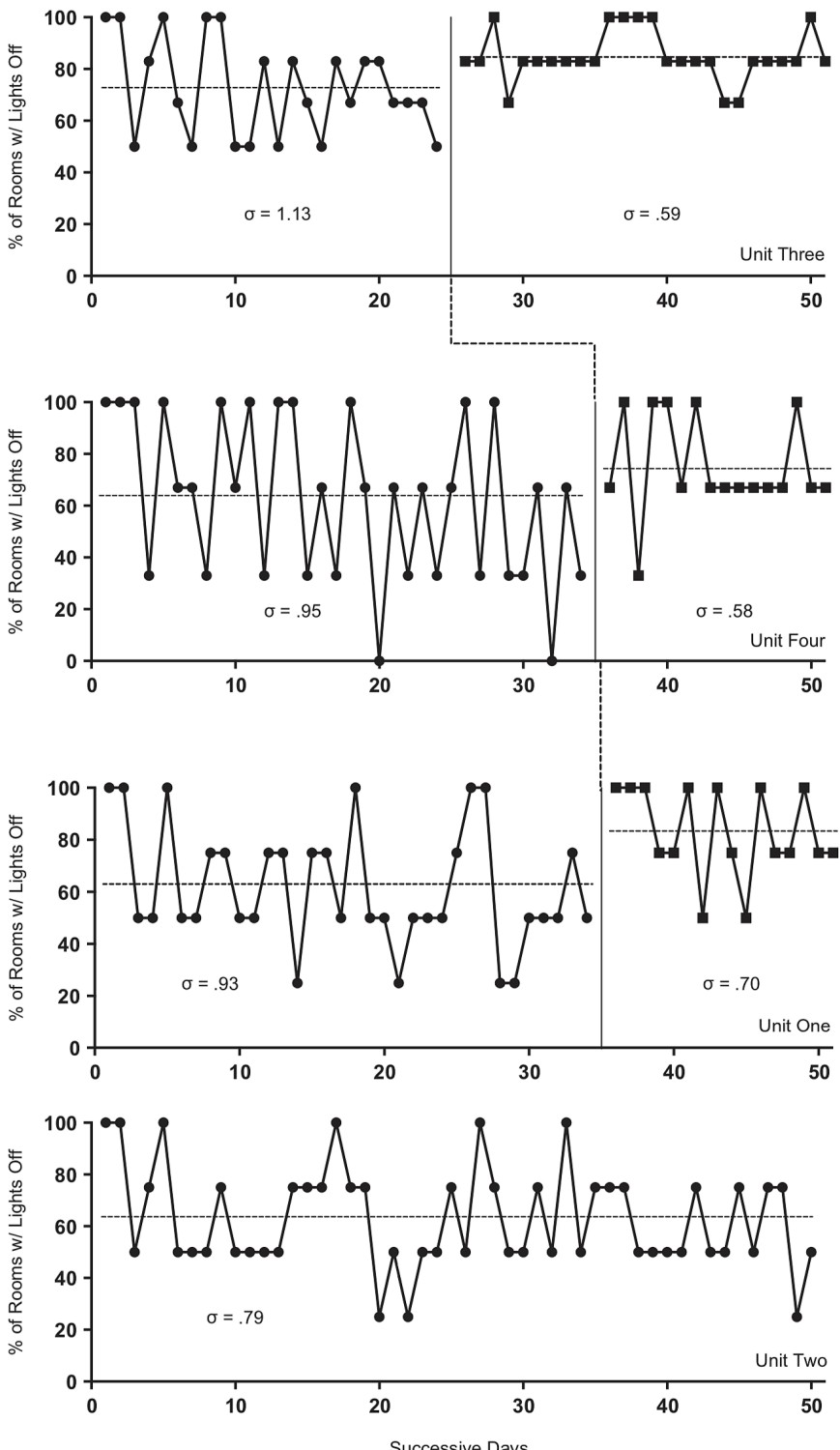

**Figure 3.** A multiple baseline design showing the percentage of rooms with the lights left on, reported by Clayton and Nesnidol [68] (Figure 3 © Taylor and Francis. Reuse not permitted).

Interrupted time-series designs are the most useful for pinpointing functional relationships between independent variables and greenhouse gas emissions. In contrast, randomized controlled trials are a less useful way to pinpoint functional relationships [65]. For example, one might evaluate the effects of a persuasive communication designed to influence the support for a community effort to reduce greenhouse gas emissions by randomly assigning people to receive or not receive such a message. However, this is a time-consuming and costly approach that may fail entirely if the persuasive message is ineffective. A better strategy would be to test the message with a series of individuals or small groups, modifying the message in light of its impact. After meaningful changes in behavior are observed, the streamlined intervention can be applied to each successive group in the context of a multiple baseline design. In essence, single-case designs encourage the ongoing refinement of interventions, in light of the immediate feedback provided by data as interventions are systematically evaluated. This feedback may guide course corrections and suggest changes to the intervention or its implementation, based upon the most recent results of its use.

Traditional randomized trials require the involvement of a relatively large number of communities, random assignment to conditions, and the standardization of the intervention across the communities. Few organizations or communities have the wherewithal to have multiple communities agree to simultaneously implement an intervention, let alone to agree to be in a control condition that never receives the intervention. These barriers are likely responsible for the lack of randomized trials in the literature we reviewed.

We believe that time-series methods can significantly improve the state of experimental research that aims to evaluate interventions to reduce emissions. These designs provide a system for enhancing an intervention, since they involve the ongoing monitoring of the targeted behavior in a way that provides feedback about what is and what is not working, thus enabling timely course corrections. This is as much a matter of the careful management of a project as it is the scientific evaluation of what works.

Although we advocate for the increased use of interrupted time-series designs, this does not preclude alternative methodologies for conducting climate change research if we can significantly increase funding for experimental research. We believe that a process of variation and selection is our best hope for evolving not only effective interventions, but more effective experimental designs. Guastaferro and Collins [69] describe the Multiphase Optimization Strategy (MOST), which involves a factorial design for assessing the relative impact of multiple intervention components. For example, one might have a community intervention consisting of three components: a school intervention to involve students in emission reduction, a household component to influence emission behavior, and a policy initiative to require organizations to audit and reduce their emissions. With multiple communities, one could randomly assign communities to receive zero, one, two, or all three of the components. Akin to a component analysis, the utility of this design is that it would not only reveal the impact of each component but also test the synergistic effects among the components. The challenge in such a design would be to obtain the resources to work in a large number of similar communities. However, given the extent to which we are failing to reduce emissions, massive increases in expenditures on experimental evaluations are imperative.

Stern [70] pointed out that most randomized trials testing emission reduction interventions have focused on affecting behaviors that occur frequently, such as daily travel. However, infrequent behaviors—such as purchasing an electric vehicle or weatherizing a house—may be more impactful. He suggests that it is difficult to experimentally evaluate the impact of strategies for affecting such behaviors because they are infrequent. We agree that it could be challenging to evaluate interventions by randomly assigning individuals to receive or not receive the intervention. We also agree that it is important to try to evaluate such strategies. Multiple baseline designs could be useful. For example, a program of incentives and advocacy to increase weatherization could be tested in a series of communities, with one community at a time being exposed to the intervention. Such a design could

enable the refinement of the strategy, such that each new community receives an intervention that has been refined based on results in prior communities.

In summary, all experimental methods—including both randomized trials and single-case designs—allow us to select increasingly effective strategies. Testing a wide variety of strategies for affecting greenhouse gas emissions, using a variety of experimental methods, will accelerate the identification of the most effective strategies for reducing GHG emissions and contribute to the prevention of further climate change.

### 4.3. The Nature of Community Interventions

Stern et al. [71] suggested design principles for any effort to reduce greenhouse gas emissions. First, prioritize high-impact actions. Second, provide sufficient financial incentives to motivate people to make major changes to their lives. Third, strongly market whatever programs are being implemented. Fourth, provide valid information from credible sources at the points of decision. Fifth, keep it simple. Sixth, provide quality assurance.

A variety of community intervention strategies should be tested. These include efforts to get policies adopted by municipalities, the implementation of policies in communities, media campaigns to influence households and organizations in the community, and school-based programs in all of the schools in the community. We suggest that the most promising interventions are those that systematically organize support for emission reduction in every sector of the community. This was the most common strategy in the community interventions conducted to affect health behavior in communities, and thus formed the foundation for our definition of a community intervention [4,6–9,72]. In the context of GHG emissions, this strategy consists of educating and engaging the leaders of every sector of the community about the need for reductions in greenhouse gas emissions, forming a cross-sector coalition of organizations that leads a community-wide process of identifying policies and practices that have been shown to have some impact on emissions, implementing those policies and practices, and creating a 'backbone' organization [73] to monitor the implementation and the impact of each strategy on its targeted outcomes. Vandenbergh and Gilligan [74] have made a strong case for the extent to which progress can be made in reducing emissions through the actions of business and other nongovernmental organizations. Community interventions can be a vehicle for increasing these actions, and for business organizations influencing governments' actions.

Among the strategies that would be offered for the leadership's consideration are policies that would increase the cost of emissions, policies that provide incentives for reductions [39], and policies that require the ongoing measurement of emissions and feedback of that information to the community as a whole, and to specific sectors of the community (e.g., households, businesses, government, transportation, and schools). Programs that could be implemented might include: (a) assistance to businesses in measuring their emissions and adopting policies and programs that help them to reduce emissions, (b) feedback and incentives to utility customers for the reduction of emissions [75], (c) school programs that educate students about reducing emissions and have the students interview their parents [76] in a way that increases parental involvement in the reduction of emissions, (d) neighborhood organizing to enhance social cohesion and promote emission reduction, and (e) enhancing social recognition for efforts to reduce emissions.

There are, admittedly, numerous types of community interventions that could be created. It is valuable to have variation in the interventions that are evaluated to investigate effectiveness over time. Certainly, one of the dimensions that could vary is the number of sectors they target. One example is the paper by Rothstein [77], which delivered a multi-component intervention through the media sector (TV), and involved both the local university and businesses in gathering data. However, it only targeted one sector with the interventions.

We believe that comprehensive multi-sector strategies will produce synergistic effects. Stern [45] has argued that our interventions need to take cognizance of the interactions of people in their many roles with energy systems—as "energy consumers, as citizens who may influence the ... regulation of

energy systems, ... as participants in organizations and institutions, and as parties affected by energy systems" (p. 41). Thus, having students interview their parents about climate change could affect parents' actions not only as a consumer but also as a citizen and a member of a work organization. In a reciprocal process, getting community organizations to adopt policies to reduce their emissions would have a salutary effect on municipal government, and getting the government to adopt policies would influence organizations.

Experimental methods—especially interrupted time-series designs—would be useful not only in assessing the overall impact on emissions in a community but also in assessing the impact of each initiative. A multiple baseline design across communities [78] could be used to assess the effects on the total emissions of communities. But such designs could also be used to assess the impact of student interviews or utility incentives on household emissions. As a form of continuous quality improvement, the latter designs would provide ongoing feedback about what was working and what needs to be abandoned or modified.

### 4.4. The Power of Behavioral Science Research

This analysis, and others that we are conducting, has revealed a surprising dearth of funding for behavioral science research on reducing GHG emissions. Ultimately, all emissions are a matter of human behavior. However, we find that far more resources are being put into technological efforts to mitigate emissions than into changing the behavior of individuals, households, organizations, or entire communities [79]. A vast body of knowledge about influencing human behavior has been accumulated thanks to experimental evaluations of treatment and prevention programs [28].

We need to put that knowledge, and the methods that produced it, to work on what may be the most important problem that humans have ever faced. To this end, the Coalition of Behavioral Science Organizations is attempting to reach out beyond the scientific community to advocate for a greatly expanded program of interdisciplinary research on reducing GHG emissions. Such a program would experimentally evaluate strategies not only for community interventions but also for getting policies adopted, and for affecting organizational and household behavior in entire populations.

### 5. Conclusions

Experimental research on community interventions to reduce greenhouse gas emissions is lacking. Without robust research designs, we risk wasting time and money on inefficient interventions. Experimental research in many other areas of behavioral science has made extraordinary progress, and we join Fischhoff [80] and McConnell [81] in advocating for a more central role for behavioral science in meeting the challenges presented by climate change. It would be a tragedy if we failed to apply these methods to what is likely to be the biggest threat to human wellbeing since the plague.

**Supplementary Materials:** The PRISMA checklist is available at https://www.preprints.org/manuscript/202006.0244/v3/download/supplementary.

**Author Contributions:** A.B. wrote the first draft of the manuscript, contributed extensive editing and feedback throughout the manuscript production, and coordinated the task force effort. A.C.B. developed the coding process together with M.J. and J.L.G., contributed to the method section and wrote the results section, coded papers, coordinated the snowball sampling, produced the figures and table, and edited all of the sections of the paper. M.J. conducted the database searches, wrote most of the method section, developed the coding process together with A.C.B. and J.L.G., and edited most parts of the manuscript. J.L.G. developed the coding process together with M.J. and A.C.B., coded papers, conducted the snowball sampling with A.C.B., and provided feedback and edits for all of the sections of the manuscript. M.J.V.R. coded papers, wrote portions of the discussion, and edited all sections of the manuscript. T.L.D. coded papers, wrote portions of the early drafts of abstract and introduction, and edited all of the sections of the manuscript. H.A.S. coded papers, and edited all of the sections of the manuscript. J.H.F. coded papers, edited all of the sections of the manuscript, re-wrote the abstract, and wrote parts of the introduction. L.W.C. provided feedback and edited all of the sections of the manuscript. All authors have read and agreed to the published version of the manuscript.

**Funding:** This research received no external funding.

**Acknowledgments:** The authors are grateful for the valuable feedback provided by Paul C. Stern on a late draft of this manuscript.

**Conflicts of Interest:** The authors declare no conflict of interest. This article is not an official position of the Behavior Analyst Certification Board.

**Author Note:** A preprint of this work can be found at https://doi.org/10.20944/preprints202006.0244.v3.

## Appendix A

Descriptions and instructions of the three stages of article coding

Stage 1

*Goal:* To identify articles that used experimental design (e.g., between-groups design, single-case design) to evaluate real life effects of an intervention on some form of GHG-emission related outcome (e.g., electricity use, food waste, gasoline consumption, CO2 emissions, etc.). We include quasi-experimental designs, but exclude computer modeling/simulations, lab experiments, etc.

*Instructions:* Indicate relevant codes by putting the number 1 in the corresponding column(s) for the appropriate code(s). If a particular code is not indicated, leave the cell blank.

*Step1:* Open the corresponding Excel spreadsheet and locate the rows assigned to you for coding (i.e., cell A3).

*Step2:* Navigate to the correct row and read the article title.

*Step3:* Indicate Code 0a (code descriptions below) if the paper is clearly irrelevant based on title. If 0a is not immediately obvious, then go to step 4. If Code 0a is indicated, continue to next article and repeat.

*Step4:* Read the abstract (you may need to double click the cell to view the whole abstract).

*Step5:* Indicate Code 0b if the paper is irrelevant based on abstract. If indicated, continue to next article and repeat. If Code 0b not indicated, continue to Step 6.

*Step6:* Indicate Codes 1–4 where relevant. Repeat for all articles.

*Step7:* Return completed template by email to volunteer coordinators.

Stage 1 Code Descriptions

*Code 0a:* Irrelevant based on title. If 0a is not immediately obvious, then read abstract and code 0b if appropriate. Code 0a should only be indicated if the title is obviously and definitively unrelated.

*Code 0b:* Irrelevant based on abstract. If Code 0a *or* 0b indicated (do not indicate both), do not indicate code 1–4.

*Code 1a:* Experimental method/design is used to evaluate real life effect of an intervention on some form of GHG-emission related outcome (ie. electricity use, food waste, food selection, co2 emission, etc). Include quasi-experimental designs, but not computer modeling simulations, lab experiments, or game theory approaches. Self-Report measures are ok at this stage, as long they are related to GHG-emission outcomes (e.g., reports of ambient home temperature before and after intervention).

*Code 1b:* Which experimental method/design? Copy and paste the relevant information directly from the abstract. If unsure, just leave blank.

*Code 2:* This is a review of literature, interventions or policies aiming to reduce GHG emissions in some way. Can for instance be retrospective longitudinal data, evaluating outcomes based on different policies.

*Code 3:* This is a meta-analysis of interventions or policies (same as Code 2, but effect-size measures are reported).

*Code 4:* Need to read the whole paper to code properly. Use this code sparingly when Code 1a is uncertain.

Stage 2

*Goal:* To determine (a) if articles indicated as Code 1a or Code 4 from Stage 1 contain an experimental evaluation of a community intervention aimed at reducing GHG emissions, and (b) to code the qualitative features of all relevant articles.

*Instructions:* Indicate relevant codes by putting the number 1 (or listing with words where indicated) in the corresponding column(s) for the appropriate code(s). If a particular code is not indicated, leave the cell blank

*Step1:* Open the corresponding Excel spreadsheet and locate the rows assigned to you for coding (i.e., cell A3).

*Step2:* Navigate to the correct row to locate the assigned article title. Retrieve the assigned article and read it.

*Step3:* If the article does not contain a community intervention aimed at reducing greenhouse gas emissions indicated Code 0 (code descriptions below). If Code 0 is indicated, continue to the next article and repeat.

*Step4:* Indicate Codes 1–8 where relevant. Repeat for all articles.

*Step5:* Return completed template by email to volunteer coordinators.

Stage 2 Code Descriptions

*Code 0:* Irrelevant based on full-text reading.
*Code 1:* Data collected directly (i.e., by automated measurement or observation).
*Code 2*: Data collected indirectly (i.e., by self-report or survey).
*Code 3:* What type of design was used (group or interrupted time series?).
*Code 4:* Indicate the exact design arrangement by selecting the cell which correctly depicts the design (O = observation, X = intervention, R = Random assignment).
*Code 5:* Indicate all dependent variables and units (e.g., electricity consumption in KwH).
*Code 6:* List all the intervention components (e.g., incentives, performance feedback, information etc).
*Code 7:* What type of community was targeted (e.g., village, town, city).
*Code 8:* Describe the overall impact of the intervention using descriptive statistics (e.g., 10% reduction in electricity) or inferential statistics (e.g., statistically significant difference between groups) presented in text.

Stage 3

*Goal:* To search the reference sections of articles indicated as Code 2 or 3 from Stage 1 for relevant titles.

*Instructions:* Paste relevant citations beside assigned articles. All new citations will undergo Stage 2 coding procedures.

*Step1:* Open the corresponding Excel spreadsheet and locate the rows assigned to you for coding (i.e., cell A3).

*Step2:* Navigate to the correct row to locate the assigned article title. Retrieve the assigned article and navigate to the reference section.

*Step3:* Scan each article in the reference section looking for article titles that suggest a relevant evaluation may be contained therein.

*Step4:* If relevant citation is found, paste it beside article title in corresponding spreadsheet.

*Step5:* Repeat for all articles assigned.

*Step6:* Return completed template to a volunteer coordinator.

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
