# Peer review of "The State of Experimental Research on Community Interventions to Reduce Greenhouse Gas Emissions—A Systematic Review"

_sustainability, doi:10.3390/su12187593_

Round 1
Reviewer 1 Report
This manuscript describes a rigurous systematic review on research in social sciences related to climate change strategies. However, its main result is not the potential generalizations of rules that could be expected from such review, but just the opposite: the almost total lack of adequately reported quantitative research on the topic. Such result is very interesting, as it is a clear wake up call on how social science research on community behaviour and climate change is described. The manuscript is well written, with reports carefully described and correctly applied, and the scarce results are well discussed. My only three suggestions are:
- Clarify the initial hypothesis and objectives at the end of the introduction. there are three reasons mentioned to explain why the research was done, but there is clear objectives derived from an initial hypothesis to be tested.
- Figure 2 seems a bit blurry.
- The discussion would benefit form a short conclusions section.
All things considered, I recommend to accept this manuscript for publication after the previous minor changes are done.
Author Response
Thank you for providing valuable feedback on our manuscript!
We added a paragraph on hypothesis and objectives.
Regarding Figure 2, we agree. Unfortunately, that is the best version we can find.
A brief Conclusions section has been added.
Reviewer 2 Report
Review of
The State of Experimental Research on Community Interventions to Reduce Greenhouse Gas Emissions - A Systematic Review
The article is a systematic review of community-level interventions on reducing GHG. The authors have done extensive work in conducting the systematic analysis and, if it were not for the lack of cases, they would have probably conducted a full meta-analysis as well. The article is written in a transparent and precise manner, making it easy for others to replicate the literature search.
The authors highlight that although there certainly are a lot of community-level initiatives aimed at reducing GHG, virtually none of them bother to measure the impact of their efforts. In cases where the impact is indeed explored, the research is mostly qualitative, thus providing no real insight into the actual effectiveness of the initiative. While the lack of cases did limit the article in terms of the conclusions that authors could make, the authors apparently took this as an opportunity to introduce their readers to possible research designs that would be suitable for evaluating community-level interventions on GHG emissions. Overall, I see this as a welcome addition to the literature that makes a very strong point – we need more robust research on how to reduce GHG emissions at the community level.
I have only one comment regarding the article: I’m not sure if you can reprint Figure 2 or Figure 3 in this journal, since all publications in Sustainability are under the CC-BY license. Please double-check this.
I wish the authors the best of luck in their ongoing and future research!
Author Response
Thank you for providing valuable feedback on our manuscript!
You raise an important question regarding licensing. Our understanding is that reusing the figures should be fine when using the text included beneath each figure, which clarifies that the figure is not included in the CC BY license that applies to the rest of the work. A note on this exception has also been added to the license notice at the end of the document.
Reviewer 3 Report
I congratulate the authors on their research goal. Identifying and disseminating the most efective strategies to mitigate climate change is pressing and critically important.
I would like to propose some points that may improve your work further:
- I believe a more robust definition of community interventions is needed: what their scope and range is, how exactly they differ from interventions targeting individuals and households. What are the conceptual reasons and mechanisms through which an intervention targeting a whole community would produce different effects? I believe this clarification is needed because even though interventions may be presented as community interventions, their implementation of experimental stimuli and data collection may be collected at the individual/ household level and then simply averaged.
- This brings me to my second point which is your search strategy. I feel that, because point 1 needs a bit more work, your search strategy seems a bit simplistic. An important characteristics of community interventions (vs. individual and household) is their group-level randomization e.g., cluster randomized trial. So I expected that one of your search terms would be "cluster random*" for instance. How about interventions randomized to different schools, buildings or organizations? Would this fall under your definition of community interventions? You maybe right that there are very few experimental field studies about community interventions but I believe your search strategy needs to be a bit improved so that the reader can be more certain that in fact that is the case.
- I would also recommend being more explicit and clear in your inclusion criteria about what your DVs are. Which behaviors are included in contributing to mitigating climate change and why. I believe an rather exhaustive list is required - and this will also help to shape your search strategy better. This also links to your section 4.1: If you are clear in your inclusion criteria, these limitations ma be less relevant.
- Please reconsider your sections 4.2, 4.3 and 4.4. The Discussion section is not typically used to present new information/ new figures. If you would like to present some insights and recommendations for future research, please reframe the text accordingly. At this point, it feels a bit like a literature review.
- Please conduct a more thorough literature review of recent reviews/ meta-analysis about behavioral science and climate change. Some potentially important recent references are missing.
- Lastly, please consider following the PRISMA guidelines http://www.prisma-statement.org
I wish the authors all the best in their research efforts on such a valuable topic.
Author Response
Thank you for providing valuable feedback on our manuscript!
- We agree and have made edits to clarify.
- Any search hits for “cluster random*” would have been included in a search that found something for “random*” (without “cluster”), which we used in our search. While our search strategy might seem simplistic, it is very inclusive. Regarding organizations/schools etc, this belongs mostly to the definition issue on community interventions, which we hope has been addressed adequately. Also, regarding the search strategy, we have conducted a separate search for interventions primarily targeting organizations, as we see this as separate from community interventions. Schools fall into the organization category as well.
- We have clarified, but not at the level of specific behaviors. The main outcome/DV is GHG emissions. Which behaviors are involved in achieving reduced GHG emissions is not something we think should be defined or limited a priori, since that could make us miss out on impactful behaviors we had not thought of. A review is likely to help map which behaviors are targeted and how effective these behaviors are at reducing GHG emissions.
- We agree that the discussion of interrupted time-series designs was too long. We have cut it down substantially. With respect to moving our discussion of multi-sector interventions to the introduction, we are reluctant to do that. We feel that it is only when the reader has been convinced that too little experimental work has been done (assuming we succeed in convincing) that we can make a compelling case for the kind of multi-sector interventions that we feel are needed. While perhaps unorthodox, we believe that our discussion section is a constructive approach to make the paper more useful to readers, considering the lack of findings.
- We have added four recent review/meta-analysis papers.
- Figure 1 is a PRISMA flowchart, and we tried to attached a PRISMA checklist to the submission, which apparently failed. The checklist is now referenced with a download link under Supplementary Material.
Round 2
Reviewer 3 Report
Dear authors,
Thank you for your efforts to improve the paper.
I have two final remarks that I believe are still relevant to improve the paper a bit more.
Firstly, the clarification of the dependent variables is needed. You mentioned that "The main outcome/DV is GHG emissions. Which behaviors are involved in achieving reduced GHG emissions is not something we think should be defined or limited a priori, since that could make us miss out on impactful behaviors we had not thought of. A review is likely to help map which behaviors are targeted and how effective these behaviors are at reducing GHG emissions."
This, however, is not totally correct because you implicitly express which behaviors (or DVs) you are interested in in your search strategy i.e.,
"TITLE-ABS-KEY(energy OR electricity OR food OR plant-based OR diet OR refriger* OR cool* OR chlorofluorocarbon OR cfc OR cryogenic* OR "heat remov*" OR "heat recov*" OR "heat exchange")"
Therefore, you do present which DVs you are interested in not in a clear and explicit way. If you thought recycling was relevant, it would have been included in the search strategy. Because you are probably not interested in recycling, it was not included. How about driving and fuel efficiency cars, wouldn't that be relevant?
There should a clear and inextricable link between your DVs and your search strategy. The terms you included in your search strategy are the DVs you are interested in. Thus, please guarantee you have covered in your search strategy all DVs relevant to GHG emissions, given that is your goal.
Secondly, and in line with my previous comment, the search strategy needs to be revised a bit. I had mentioned some need for revision in my previous comments. I know this may be a bit troublesome but as a reviewer my role is to guarantee as much as possible that your procedure can attest the conclusions you made. With the current search strategy, that is not completely clear.
My main note is your use of " TITLE-ABS-KEY(community OR communities) AND..." By using AND here, you are eliminating any paper that does not use the explicit term community(ies).
But by your own definition of community intervention, papers that you are trying to identify may not use the term community: "Examples of community intervention strategies would include organizing neighborhoods to reduce emissions, getting a city council to adopt ordinances that would affect emissions, or attempts to influence local business organizations to reduce their emissions."
Therefore, you search terms should be expanded to include further, in addition to community, for instance, neighborhood, schools, organizations, municipalities, large-scale etc or similar terms that may capture the idea of community intervention without explicitly using that term. I would still advise to include "cluster" in the part of the search about research design because some papers only use the expression "cluster trial", and not randomized, even though it may be randomized.
Here is an example of a study that probably could interested you:
All the best of luck and I will be satisfied with these two changes.
Thank you.
